# Validation, Reliability, and Responsiveness Outcomes of Kinematic Assessment with an RGB-D Camera to Analyze Movement in Subacute and Chronic Low Back Pain

**DOI:** 10.3390/s20030689

**Published:** 2020-01-27

**Authors:** Manuel Trinidad-Fernández, David Beckwée, Antonio Cuesta-Vargas, Manuel González-Sánchez, Francisco-Angel Moreno, Javier González-Jiménez, Erika Joos, Peter Vaes

**Affiliations:** 1Rehabilitation Sciences Research Department, Vrije Universiteit Brussel, 1090 Brussels, Belgium; m.trinidad@uma.es (M.T.-F.); david.beckwee@vub.be (D.B.); pvaes@vub.be (P.V.); 2Clinimetric Group F-14, Department of Physiotherapy, Biomedical Research Institute of Malaga (IBIMA), University of Malaga, 29010 Málaga, Spain; 3Department of Rehabilitation Sciences and Physiotherapy, University of Antwerp, 2000 Antwerp, Belgium; 4School of Clinical Science, Faculty of Health Science, Queensland University Technology, Brisbane 4072, Australia; 5Machine Perception and Intelligent Robotics Group (MAPIR), Dept. of System Engineering and Automation, Biomedical Research Institute of Malaga (IBIMA), University of Malaga, 29071 Málaga, Spain; famoreno@uma.es (F.-A.M.); javiergonzalez@uma.es (J.G.-J.); 6Physical Medicine & Rehabilitation Department, UZ Brussel, 1090 Brussels, Belgium

**Keywords:** depth camera, motion capture system, low back pain, functional test, validation, responsiveness, reliability

## Abstract

Background: The RGB-D camera is an alternative to asses kinematics in order to obtain objective measurements of functional limitations. The aim of this study is to analyze the validity, reliability, and responsiveness of the motion capture depth camera in sub-acute and chronic low back pain patients. Methods: Thirty subjects (18–65 years) with non-specific lumbar pain were screened 6 weeks following an episode. RGB-D camera measurements were compared with an inertial measurement unit. Functional tests included climbing stairs, bending, reaching sock, lie-to-sit, sit-to-stand, and timed up-and-go. Subjects performed the maximum number of repetitions during 30 s. Validity was analyzed using Spearman’s correlation, reliability of repetitions was calculated by the intraclass correlation coefficient and the standard error of measurement, and receiver operating characteristic curves were calculated to assess the responsiveness. Results: The kinematic analysis obtained variable results according to the test. The time variable had good values in the validity and reliability of all tests (r = 0.93–1.00, (intraclass correlation coefficient (ICC) = 0.62–0.93). Regarding kinematics, the best results were obtained in bending test, sock test, and sit-to-stand test (r = 0.53–0.80, ICC = 0.64–0.83, area under the curve (AUC) = 0.55–84). Conclusion: Functional tasks, such as bending, sit-to-stand, reaching, and putting on sock, assessed with the RGB-D camera, revealed acceptable validity, reliability, and responsiveness in the assessment of patients with low back pain (LBP). Trial registration: ClinicalTrials.gov NCT03293095 “Functional Task Kinematic in Musculoskeletal Pathology” 26 September 2017

## 1. Introduction

Low back pain (LBP) has a high prevalence during life, and the costs for society in Europe are extremely high [1,2,3]. LBP is caused by the interaction of physical disfunction, anatomical variables, and psychosocial comorbidities, e.g., sedentary life or depression [4,5]. In up to 90% of all patients, low back pain is considered non-specific, since no specific cause can be detected [6].

The diagnosis of non-specific LBP is a challenge due to poor correlation between findings in spinal imaging and symptoms [7,8]. In addition, it may not be possible to define a precise cause of lower back symptoms [9]. The functional limitations must be analyzed with objective tests, because questionnaires and visual assessment by clinicians can have an important subjective load in the movement examination [10,11]. 

Functional tests are useful in the assessment of LBP patients [10]. Therefore, it is necessary to investigate new objective assessment tools in order to use these functional tests. One of the potential options are RGB-D cameras, or depth cameras, because they can create a real-time human motion capture system. 

Depth cameras are easy to use, low-cost, have been used in several previous kinematic studies, and can easily translate from the lab to everyday clinical environments [12,13,14,15]. In addition, it is not necessary to attach devices on patients in a way that can possibly limit movement or create discomfort. Before implementation in the clinical practice, it is important to know the psychometric properties of outcome measures, such as reliability, validity, or responsiveness, to select the best suited assessment [16]. Previous studies have shown that depth camera technology can be a good option to analyze movement. Moreno et al. (2017) showed good results for the reliability of the camera (ICC = 0.844–0.813) and moderate to excellent internal validity with an inertial measurement unit (IMU) in the multidirectional reach test (r = 0.59–0.98) [15]. Currently available IMUs are the optimal method to instrument functional tests in order to measure [17], but the accuracy of these measurements relates to whether the markers are placed on the skin and create an interaction with the patient that can be avoided with a RGB-D camera [18,19]. On the other hand, conventional 3D video motion analysis systems, such as the VICON system, are expensive and limiting to use in clinical practice or tele-rehabilitation programs, unlike the ease of the depth cameras [20,21].

The measurement properties of this type of human motion capture system must be analyzed in order to justify its use in a systematic assessment in LBP patients. Lee et al. (2011) showed the importance of kinematic assessment in LBP patients with an inertial sensor [22], but there are no studies that measure, for example, the kinematics of trunk flexion in LBP using an RGB-D camera. 

The aim of this study was to present results about internal validity, reliability, and external responsiveness outcomes of the motion capture system with an RGB-D camera in subacute and chronic LBP patients through kinematic analysis of trunk flexion. The internal validity was checked, comparing the camera with another kinematic tool, in this case, an IMU, the reliability was analyzed, with several repetitions screened by the camera in order to show the consistency, and the external responsiveness was classified with the Global Perceived Effect (GPE) scale in order to reflect changes of health.

## 2. Material and Methods

### 2.1. Design

This study is part of a longitudinal prospective study. This study has been registered with ClinicalTrials.gov (NCT03293095).

### 2.2. Participants and Intervention

Participation in the study was offered to patients who met the criteria at the end of the medical consultation by a physician.

Participants were included based on criteria, such as between 18 and 65 years, non-specific lumbar pain after 6 weeks following an episode, pain in the lumbosacral region with or without radiating pain in the gluteal region and the upper leg, and no clear increase in activity level and reduction in participation restrictions after 3 weeks [6,23]. 

Participants with low back pain as a result of a specific spinal disease, infection, presence of a tumor, osteoporosis, fracture, inflammatory disorder, or cauda equina syndrome were excluded^6^. In addition, they were excluded if they had a hip arthroplasty in the last 6 months, participated in a study with an experimental treatment during the recruitment or the study, had a severe cardiovascular disease (Category D) according to the Senior Europeans (SENIEUR) protocol [24], had alterations in the participant’s cognition that did not allow us to understand the order of the research, or were pregnant.

The measurement was performed twice: The initial measurement and another 1 month later, in order to study responsiveness. For responsiveness, the changes across the time period between the two measurements were tested (4 weeks natural course). All patients followed a conservative low back pain exercise program.

### 2.3. Setting

The study was carried out in the UZ Brussel Hospital in Brussels, Belgium. Patients were recruited from the physical medicine department of the hospital from January 2019 to June 2019.

### 2.4. Ethical Considerations

The Medical Ethics Committee of the UZ Brussel Hospital approved this study (Nr. 2018/366). The guidelines for good clinical practice (GCP), the principles of Declaration of Helsinki, and the Belgian Law of 7 May 2004 related to experiments on humans were followed. All subjects gave their informed consent for inclusion before they participated in the study. The subjects received an informed consent with information about the study that was the signed by the patient. The patient was free to stop and leave the study at any time.

### 2.5. Motion Capture RGB-D Camera System and Inertial Measurement Unit

The motion capture RGB-D camera Xtion Pro (ASUS, Taipei, Taiwan) was used in the study. The distance between the camera and the participant was set approximately at 2.5 m. The camera was placed between 40–45°, with respect to the direction of movement, and at a height of 90 cm from the floor (Figure 1).

The IMU MP67B (InvenSense, San Jose, USA) from an iPhone6s (Apple Inc., Cupertino, CA, USA) collected the information about mobility angle and acceleration along three axes with the gyroscope, accelerometer, and magnetometer. The IMU showed high accuracy in a medically acceptable limit (±5°), and the angular velocity noise level was 0.09°/s [25]. This methodology, based on an IMU from a smartphone, was validated previously using the timed up-and-go (TUG) test [26]. The smartphone was placed at the level of the thorax over the sternum inside a specific belt around the thorax (Figure 2). The SensorLog® 2.2v app from Apple App Store processed the sensor data using the Core Location and Core Motion frameworks from the iPhone. The recording rate was set at 100 Hz. A 3D coordinate references system for both instruments was used (Figure 2).

### 2.6. Functional Tests

Functional tests are based on physical assessment used in LBP patients and inspired by frequently impaired daily activities as described by the patients [27,28,29,30]. (Figure 3):(a)Modified stairs climbing test (stairs test): Subject had to climb two-steps stairs without assistance by placing one foot on each step (height and depth of each step was 15 × 30 cm) [27].(b)Bending test: A pen was placed on the floor in front of the subject. The subject was asked to bend forward from the hips and pick up the pen without assistance [27].(c)Reaching test: Subject facing a shelf placed at patient’s head height +15%. Patient was instructed to place a pen on the shelf without help or assistance [27].(d)Sock test: Subject had to put on his sock on the dominant foot sitting without help or assistance. The chair had 44 cm sitting height [27].(e)Lie-to-sit test: Patient had to perform the lying-to-sit transition [28]. Starting from a supine position, the patient was asked to turn on his side and then sit using his arm, while the legs were lowered at the side of the table.(f)Sit-to-stand test (STS test): A chair with a 44 cm sitting height was used. The patient was instructed to stand up and sit down from the chair without using hands or assistance [29].(g)TUG test: The patient started seated on a chair (44 cm seating height) and was asked to get up and walk until reaching a cone at a 3 m distance from the chair, turn around it, return it to the chair, and sit down again. Patients walked as fast as possible without running [30].

All the tests were standardized in order to improve accuracy in the analysis.

### 2.7. Questionnaires

The validated version of questionnaires in French and Dutch were used in order to describe the sample. The Roland–Morris Disability Questionnaire (RMQ) was used for low back disabilities [31,32] and the EuroQoL-5D-VAS [33,34] and SF-12 questionnaires [35] were used for quality of life and health. 

A GPE scale was used to collect the overall measure of change during the month between the measurements [36]. The scale had 7 items that ranged from 1 (“very much improved”) through to 4 (“no change”) to 7 (“very much worse”) [36].

### 2.8. Measurement Procedure 

The approximate time was 60 min per test. The measurement was divided into three parts: filling in the questionnaires, preparing the participant, and performing the functional tasks.

The smartphone was placed on the patient using a belt at the level of the thorax, and the motion capture area of the depth camera measurement was shown to the patient on the computer screen. Participants watched a video of each test before each measurement, and the rater gave them a standardized instruction. Hereby, they could familiarize each test before the data collection. After performing the starting position that allowed the body recognition by the camera, participants performed the maximum number of repetitions during 30 s for each functional task and then a rest of 120 s was allowed following each test, in order to prevent fatigue. Three repetitions of the TUG were recorded. 

The participant was in a static position at the beginning and end of each test for 10 s in order to improve the synchronization of both data sets in the data processing.

### 2.9. Variables

Displacement (degrees), time (seconds), velocity (m/s), and acceleration (m/s^2^) were obtained from the three assessment tools. The flexion-extension trunk displacement was calculated directly from the data and represented by the pitch angle and the anteroposterior acceleration by the acceleration in Z, as shown in Figure 2. Velocity and acceleration were calculated indirectly based on the following formulas: “velocity = displacement/time” and “acceleration = velocity/time”. The outcomes were extracted from the interval of movement between control points. Functional tests were marked with two control points: the starting position (A) and the ending position (B) of each test. Therefore, the A→B interval was measured. The control points in the TUG test were: the starting point (A); the stand-up position (B); reaching the turning point (C); the point immediately before the participant starts to sit down (D); and the return to the starting point (E). Consequently, A–B, B–C, C–D, and D–E intervals were measured. Following this procedure, the kinematic pattern of each test and the selected intervals were obtained (Figure 3).

### 2.10. Data Recording and Processing

Anthropometric characteristics (age, weight, height, and body mass index) were recorded for each participant. Kinematic data was correlated with the timestamp provided by each tool. A timestamp is a sequence of characters giving the date and time of day. The synchronization between both devices (camera and IMU) was made with the time-stamp data from both devices and the 10 s before and after the test by a researcher.

Software libraries OpenNI2 and NiTE2 were used to extract the information from the RGB-D camera and create a virtual skeleton representation with the location of the skeletal joints (Figure 2). The representation was captured when the patient was in front of the camera and performed the starting position with the upper limbs raised sideward. The software, MRPT, was developed for a previous study [15] and has been released publicly as part of the open-source software library [37].

The parameterization to calculate the patient’s movement delivered inclination angles and angular speed between the skeletal joints. The 3D positions that corresponded to the “neck” and “torso” joint labels (Figure 2) were used to calculate the angle between them as the trunk flexion, because it coincided with the movement of the center of mass. This coincided with measuring body motion at the T7 level. The inertial measurement unit was placed over the chest at the same level. The smartphone’s orientation and the dimension of space were measured as follows: flexion–extension (α, pitch angle): rotation axis was Y, with positive data indicating flexion, and negative values indicating extension [15]. Rotation (β, yaw angle): the rotation axis was X, where positive data indicated right rotation, while negative values indicated left rotation [15]. Finally, inclination (γ, roll angle): the inclination axis was Z, where positive data indicated right inclination, while negative values indicated left inclination [15].

In the case of the depth camera, let **P***_N_* = (*X_N_*,*Y_N_*,*Z_N_*) and **P***_T_* = (*X_T_*,*Y_T_*,*Z_T_*) be the 3D spatial coordinates of the neck and torso joints as measured by the range camera, respectively [15]. The equivalent flexion–extension (α) angles can then be computed as [15]:$$\alpha =\arctan\left(\frac{X_{N}-X_{T}}{Y_{N}-Y_{T}}\right)$$

Mean and standard deviation (SD) were calculated for time, displacement, velocity, and acceleration.

### 2.11. Statistical Analysis

The internal validity, reliability, and external responsiveness were analyzed using the previously outlined variables. The third repetition was chosen for the validity and responsiveness analysis, and the first three repetitions were chosen for the reliability analysis. There repetitions were chosen in order to avoid fatigue. If the participant was not able to perform three repetitions due to the severity of the condition, the third one repetition was calculated as an average of the first two. In addition, a descriptive analysis was performed on each variable from the kinematic devices and questionnaires, and the mean and SDs were included. 

Internal validity was calculated by the correlation between the measurements of the RGB-D camera and the IMU using a parametric test, Pearson correlation or non-parametric test, or Spearman correlation (r), according to the data distribution by the Kolmogorov–Smirnov test previously used [38]. The correlation values were classified into three categories: poor (*r* ≤ 0.49), moderate (*r* = 0.50–0.74), and strong (*r* ≥ 0.75) [38]. A Bland–Altman plot was created for those tests with moderate or strong correlation in the kinematic variables (displacement, velocity, and acceleration) to show the agreement of the measure tools. 

The reliability was measured as a way of monitoring the measurements by the intraclass correlation coefficient (ICC) two-way random-effects model 2.1, 95% CI, and the standard error of measurement (SEM). The reliability results were classified into these categories: poor (ICC ≤ 0.49), moderate (ICC = 0.50–0.74), good (ICC = 0.75–0.89), and excellent (ICC ≥ 0.90) [39].

The area under the curve (AUC) of the receiver operating characteristic (ROC) curves was the chosen method to quantify the external responsiveness [40,41]. The external criteria to classify the patient for external responsiveness analysis was the global perceived effect scale. Two categories were created in order to obtain a dichotomic variable. The categories from 1 to 3 (“very much improved” to “a little improved”) were classified as “improved”. The categories from 4 to 7 (“no change” to “very much worse”) were classified as “nonimproved” [36]. The levels of external responsiveness were classified according to the AUC in low (0.50–0.70), moderate-to-high precision (0.70–0.90), and high precision (0.90) [40].

Data analysis was conducted by an external, blinded, and expert researcher. All analyses were done using SPSS version 22 software (SPSS Inc., Chicago, IL, USA).

## 3. Results

Thirty subjects participated in the initial measurement. A total of 23% (n = 7) of the patients measured in the first phase did not complete the study because they did not attend the second measurement 1 month later. The mean of the anthropometric characteristics and the score of the questionnaires were calculated (Table 1). The mean and standard deviations of the kinematic variables and the repetitions were determined in each test (Table 2).

Regarding the lie-to-sit (LTS) test, there were patients who did not perform more than two repetitions due to the severity level of complaints. Therefore, this test was completed only by 27 subjects in the first measurement and 19 subjects in the second.

Of the participants whom completed the study, the results of the GPE scale were: 1—very much improved 0%, 2—much improved 13%, 3—a little improved 34.8%, 4—no change 39.1%, 5—a little deterioration 4.3%, 6—much worse 4.3%, and 7—very much worse 4.3%. Therefore, 47.8% of the sample was classified in the category of "improved" and 52.2% was classified in the category of "nonimproved".

The results of internal validity, reliability, SEM, and responsiveness are shown in Table 3. The time variable had excellent values in the validity of all tests (r = 0.93–1.00) and the reliability was between moderate and excellent (ICC = 0.62–0.93). Regarding displacement, velocity, and acceleration, the tests with moderate to strong internal validity and reliability results were the bending test (r = 0.53–0.99, ICC = 0.75–0.93), STS test (r = 0.59–0.92, ICC = 0.64–0.92), and sock test (r = 0.53–0.99, ICC = 0.64–0.83). The best responsiveness results were the STS test (AUC = 0.64–0.85) and the stairs test (AUC = 0.60–0.84).

The data used in this study are available in Appendix A.

The Bland–Altman graphs for agreement illustrate this for each of these tests (Figure 4, Figure 5 and Figure 6).

## 4. Discussion

The aim of this study was to analyze the internal validity, reliability, and external responsiveness of a human movement capture system using an RGB-D camera or depth camera. Following the recommendation of Clark et al. (2019), the information extracted from the chosen angle was carefully selected and validated for these functional tests and this type of patient, so it is expected that future studies will check the clinical contribution of this assessment [17]. Broadly, the time variable obtained the results closer to 1, while the other measurement properties, such as displacement, velocity, and acceleration, showed an internal validity between poor to moderate (r = −0.12–0.80), a reliability between poor and excellent (ICC = −0.01–0.93), and external responsiveness between low and moderate (AUC = 0.55–0.84). 

### 4.1. Six Functional Tests

The results were different and irregular depending on the test, as they were in previous studies. For example, Moreno et al. (2017) and Mentiplay et al. (2018) showed good results in both properties (r = 0.59–0.98, ICC = 0.81–0.84 and r = 0.76–0.96, ICC = 0.91–0.96, respectively) [15,42]. On the other hand, another very similar study showed poor results in most of the correlations (r < 0.4) [14]. There were three tests in the present study that had better results, reaching minimal quality norms, according to the classification shown in the statistical analysis of the STS test, bending test, and sock test. The three tests have moderate results in common in terms of internal validity (r = 0.53–0.80) and moderate to excellent reliability (ICC = 0.64–0.93). In addition, they are tests where the trunk flexion was greater than 15°, except for the LTS test. The LTS test had poor results (r = 0.09–0.24, ICC = 0.16–0.48), which may be due to the complexity of the task and the overlapping joint points [17].

The STS test obtained good correlation data (r = 0.59–0.73), reliability (ICC = 0.64–0.75), and responsiveness (AUC = 0.64–0.77). The STS test is one of the functional tests that evaluates the strength of the most used lower limbs [43] and it has already been used as a test to study the reliability of the depth chamber, but never in LBP patients [44]. Galna et al. (2014) and Matthews et al. (2019) compared a RGB-D camera with an active motion capture system with markers in patients with Parkinson’s disease (r = 0.99, ICC = 0.98) [20] and healthy patients (ICC = 0.97–0.98, mean absolute error = 1.7–2.8) [44], respectively. They obtained good results but they measured the linear displacement of the head [20] and the center of mass [44], unlike this study, which took the angle formed by the trunk flexion. Regarding the agreement, the data were more compact around the average compared to the other tests and had lower limits (SD = 12.74, 4.21, 0.39) than the bending test. Mentiplay et al. (2018) obtained an agreement with better limits to these tests (SD = 1.96) by measuring lumbar flexion during a single leg squat [42]. The difference in the agreement may be due to the different criterion validity used in both studies. Due to these results and other results in previous studies, the STS test captured by a depth camera can be a valid clinical tool to analyze the functionality of the patient, in addition to being an easy test to analyze [45].

The bending test obtained acceptable results in validity and responsiveness (r = 0.53–0.80, AUC = 0.55–0.84) and good results in reliability (ICC = 0.75–0.83). These results are worse than those achieved by Moreno et al. (2017), with a balance test based on trunk flexion from standing, but with the same displacement and velocity methodology (r = 0.71–0.87, ICC = 0.90–0.91) [15]. On the other hand, Galna et al. (2014) obtained worse results in a trunk flexion from standing (r = 0.49, ICC = 0.16) [20]. This disparity in the results perhaps may be influenced by the amplitude of the flexion, since, in the study by Galna et al. (2014), subjects flexed the trunk with a mean of 12°; a large difference compared to this study (66°) and the Moreno et al. (2017) study (55°) [15]. These results on displacement are consistent with what has been previously commented. The tests in this study that had an average flexion of less than 16° had worse reliability and validity data.

Regarding the sock test, no other studies have been found that analyze the kinematics of this test. This test obtained worse results than the previous mentioned tests, and there was a greater dispersion between points in the Brand–Altman plot. In addition, the difference between the means in this test was greater than other tests (IMU = 16.52°, RGB-D Camera = 30.95°). Therefore, the acceptability of this data could not be considered with equal firmness in the sock test and the bending test or the STS test. New studies on the kinematics of this test should be more precise in order to validate this test and prove if it can be useful for assessing and classifying patients [46].

Finally, in terms of responsiveness, the best results in velocity and acceleration were obtained in the bending test, STS test, and stairs test (AUC = 0.71–0.84). The data on the displacement in all the tests was poor (AUC = 0.55–0.77), except in the stairs test, but its low results in reliability and validity made its recommendation difficult as a test in front of a depth camera. The good data in the STS tests and bending tests in velocity and acceleration (AUC = 0.72–0.84), together with the other reliability and validity analysis, showed the relevance of the kinematic variables for the assessment of the individual, as also shown by Galán–Mercant et al. (2016) [47]. Objective tools are needed that offer information that the human being is not able to analyze visually.

### 4.2. TUG Test

The results in the TUG test were quite poor compared to the results of the other functional tests (Table 3). The acceleration variable in the first two phases showed acceptable validity (r = 0.60–0.66), but the reliability varied between low and moderate (ICC = 0.08–0.57). Other studies have used RGB-D cameras to examine the TUG test and found that the length of the first step may provide important clinical information [48]. We did not examine this outcome measure in our cohort. Regarding the study by Moreno et al. (2017), they obtained moderate but better results compared to this study in trunk displacement and velocity in the first and last intervals (r = 0.64–0.67, r = 0.58–0.79) and poor results in the intermediate intervals (r < 0.10, r < 0.01) [15]. One possible explanation for this difference is related to technics, namely that the subject took the risk of getting too close to the camera and there may be an exceptional loss of recorded signal due to the working range area [49], and another explanation is the problem of overlapping joint points in the turn of the TUG test. The overlapping joint problem is indeed a problem of the depth camera and it is a point to take into account [17].

Regarding reliability, Moreno et al. (2017) obtained better results in displacement (ICC = 0.83–0.84) and speed (ICC = 0.81–0.83) [15], as did Vernon et al. (2014) (ICC = 0.73, 0.93, respectively) [50], although this study only focused on assessing the first and last phase of the TUG test, where trunk flexion is predominant in getting up and sitting in the chair. It is important to reflect on why the results obtained in this study in the TUG test were not consistent with those observed in previous studies, because the TUG test did not show good results under the present circumstances. Perhaps, future studies should take into account the complexity of the test and positioning of the cameras because it is necessary to investigate the validity of a depth camera with another different perspective. 

### 4.3. Limitations

The limitations of the depth camera in this study should not be overlooked. An important limitation was that there were patients who passed the inclusion criteria but could not perform the three repetitions that were requested in the LTS test. This occurred mainly because the level of severity of the complaints was so high that some patients could not perform several repetitions. 

The poor–moderate correlation in most of the kinematic variables could be due to the criterion validity chosen for this study. The IMU and the RGB-D camera could measure the trunk flexion, although they did not share the same reference system, since the IMU was attached to the trunk and the camera took a global representation of the body. Despite this, a previous study already correlated them satisfactorily [15]. It is also a good tool to assess the kinematics with great applicability [26,51] and the data from previous studies that used different gold standard show results, similar to that commented above.

The systematic review of Papi et al. (2018) recommended performing the kinematic analysis of the whole body in patients with LBP, not limiting the analysis to the lumbar region [10]. On the other hand, Clark et al. (2019) says that the angles obtained from trunk flexion can have high precision through precalibration [17], so an advantage of the depth camera is that it does not need prior calibration like other motion capture systems [44]. This study of the RGB-D camera and LBP, collecting the information from the lumbar region and trunk movement and being the location of the center of mass, is a relevant motion descriptor as a kinematic point [52], and this reference has been chosen several times in the literature [26,53]. 

## 5. Conclusions

The RGB-D camera used to assess functional tests can be a valid tool depending on the type of test to be analyzed. Kinematics analyzed during the STS test, bending test, and sock test reached validity, reliability, and responsiveness measures from moderate to good, and this procedure could have potential in the assessment of movement or motor control in patients with LBP. Therefore, large movements are detected with acceptable reliability and validity, although smaller or more precise movements must be further analyzed in future studies to improve the registration and analysis protocol.

## Figures and Tables

**Figure 1 sensors-20-00689-f001:**
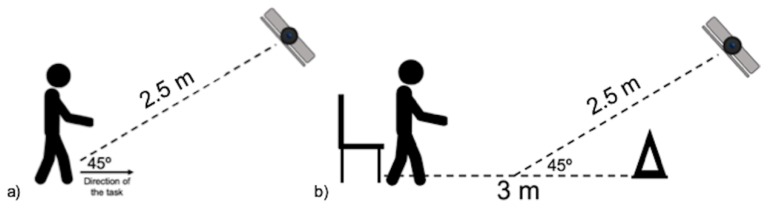
Experimental setup for (**a**) the functional tests and (**b**) the timed up-and-go test.

**Figure 2 sensors-20-00689-f002:**
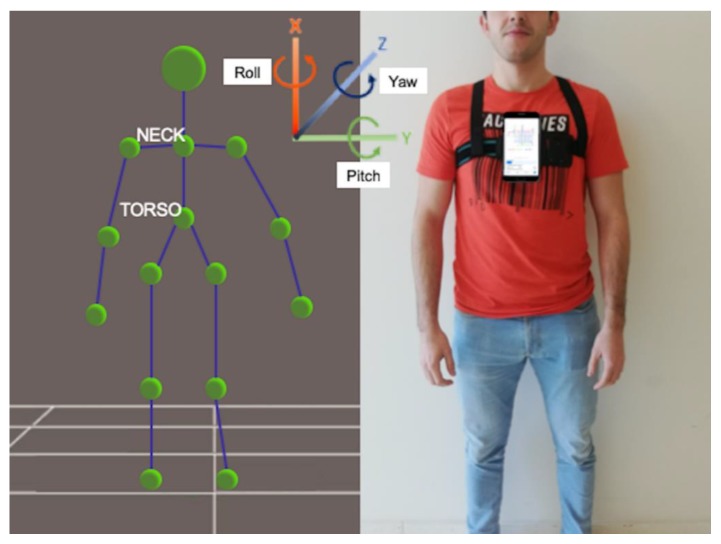
Joints information collected by the camera and 3D reference system of the camera and the inertial measurement unit.

**Figure 3 sensors-20-00689-f003:**
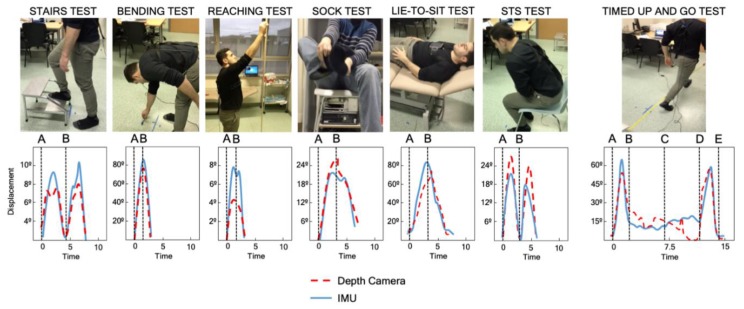
Functional tests and timed up-and-go test. Examples of kinematic pattern of each performance and control points.

**Figure 4 sensors-20-00689-f004:**
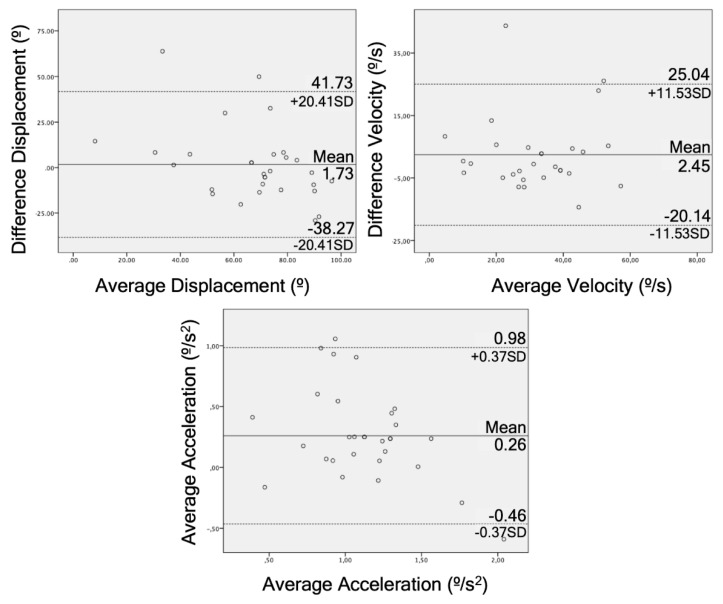
Bland–Altman plots for displacement, velocity, and acceleration in the bending test comparing the RGB-D camera and IMU. The lines represent the mean of the differences and limits of agreement.

**Figure 5 sensors-20-00689-f005:**
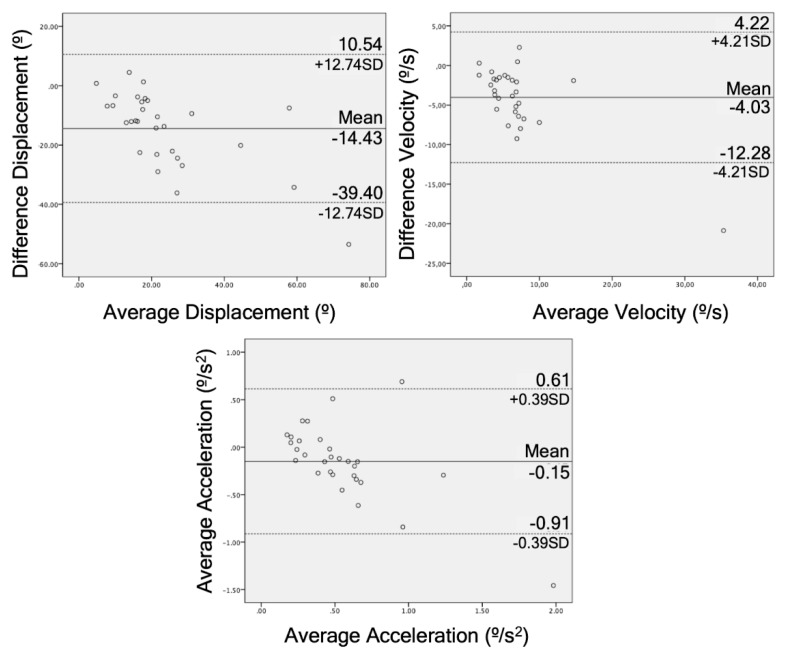
Bland–Altman plots for displacement, velocity, and acceleration in the STS test comparing the RGB-D camera and IMU. The lines represent the mean of the differences and limits of agreement.

**Figure 6 sensors-20-00689-f006:**
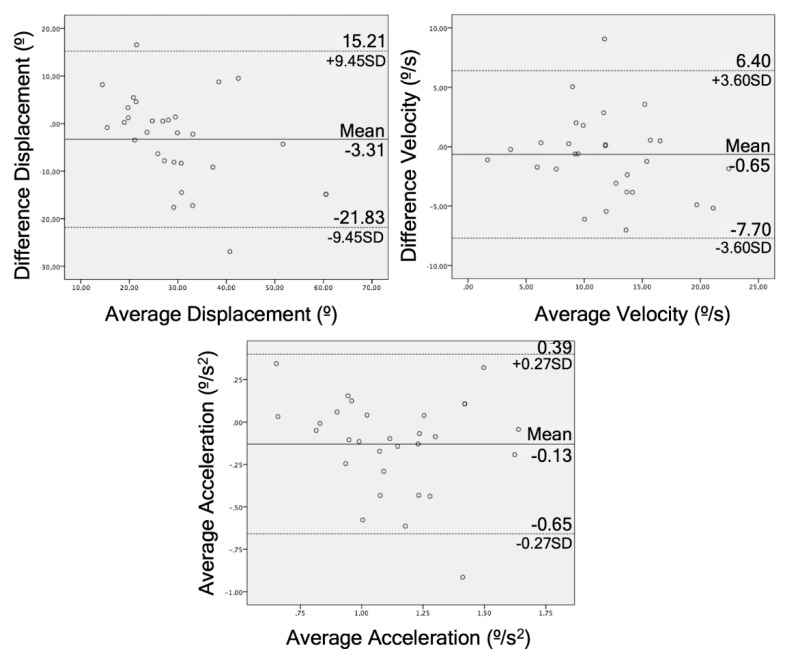
Bland–Altman plots for displacement, velocity, and acceleration in the sock test comparing the RGB-D camera and IMU. The lines represent the mean of the differences and limits of agreement.

**Table 1 sensors-20-00689-t001:** Anthropometric and questionnaire data of the sample (baseline and 1 month later). Body mass index (BMI); Roland-Morris questionnaire (RMQ).

	Pre-			Post-		
	Men (n = 15)	Women (n = 15)	TOTAL (n = 30)	Men (n = 13)	Women (n = 10)	TOTAL (n = 23)
**Age**	47.73 (12.84)	44.00 (13.03)	45.87 (12.85)	51.00 (10.04)	44.70 (15.39)	48.26 (12.73)
**Height**	176.67 (5.64)	166.47 (8.26)	171.57 (8.67)	176.77 (6.07)	163.90 (8.63)	171.17 (9.65)
**Weight**	79.13 (7.65)	65.93 (13.23)	72.53 (12.56)	80.77 (6.28)	67.50 (12.67)	75.00 (11.51)
**BMI**	25.43 (2.99)	23.93 (5.27)	24.68 (4.28)	25.93 (2.73)	25.26 (5.15)	25.64 (3.87)
**RMQ**	13.00 (5.74)	10.26 (5.95)	11.63 (5.91)	11.15 (6.24)	7.90 (4.95)	9.73 (5.83)
**EuroQoL-5D**	0.43 (0.24)	0.54 (0.22)	0.48 (0.23)	0.51 (0.20)	0.62 (0.23)	0.56 (0.21)
**EuroQoL-VAS**	59.33 (13.74)	54.46 (17.23)	56.90 (15.51)	53.46 (17.70)	67.30 (13.96)	59.47 (17.32)
**SF-12 Physical**	36.33 (8.37)	37.35 (8.10)	36.84 (8.11)	37.17 (9.42)	38.03 (6.47)	37.54 (8.11)
**SF-12 Mental**	40.27 (7.47)	39.49 (6.86)	39.88 (7.06)	40.63 (5.92)	42.54 (9.72)	41.46 (7.66)

**Table 2 sensors-20-00689-t002:** Mean (standard deviation) of repetitions and the outcomes (time, displacement, velocity, and acceleration) from the kinematic tools during the functional tests (baseline and 1 month later). Inertial measurement unit (IMU); RGB-D camera (CAM); lie-to-sit (LTS); sit-to-stand (STS); timed up-and-go (TUG).

			Time(s)			Displacement(°)			Velocity(°/s)			Acceleration(°/s^2^)		
	Repetitions Pre	Repetitions Post	IMU	CAM Pre	CAM Post	IMU	CAM Pre	CAM Post	IMU	CAM Pre	CAM Post	IMU	CAM Pre	CAM Post
**Stairs**	5.23 (1.79)	4.60 (1.69)	3.21 (1.10)	3.21 (1.11)	3.51 (1.17)	9.76 (3.42)	13.71 (10.46)	20.87 (22.46)	3.20 (1.16)	4.32 (2.94)	6.22 (6.43)	0.42 (0.09)	0.75 (0.25)	0.93 (0.34)
**Bending**	7.23 (3.01)	7.26 (2.94)	2.50 (1.67)	2.51 (1.67)	2.67 (2.38)	68.25 (18.81)	66.51 (26.62)	74.86 (21.23)	34.49 (16.87)	32.03 (16.57)	38.00 (17.91)	1.25 (0.31)	0.98 (0.45)	1.02 (0.41)
**Reaching**	9.40 (2.97)	9.13 (2.52)	1.85 (0.93)	1.84 (0.92)	1.83 (0.79)	6.51 (3.59)	4.98 (3.38)	4.89 (4.43)	4.00 (2.07)	3.08 (2.24)	3.07 (2.98)	0.12 (0.05)	0.29 (0.25)	0.32 (0.27)
**Sock**	3.66 (1.26)	3.91 (1.34)	3.87 (1.89)	3.87 (1.90)	4.35 (2.19)	16.52 (12.43)	30.95 (20.61)	32.92 (19.95)	4.84 (4.53)	8.86 (7.65)	9.15 (7.89)	0.47 (0.28)	0.62 (0.51)	0.58 (0.38)
**LTS**	3.07 (1.07)	3.63 (0.83)	4.87 (1.24)	4.88 (1.24)	4.45 (1.26)	91.75 (35.61)	93.23 (12.86)	94.26 (45.88)	20.25 (5.50)	19.33 (7.15)	21.77 (9.80)	1.31 (0.27)	2.06 (0.88)	2.32 (1.57)
**STS**	6.26 (1.98)	6.47 (2.31)	3.25 (4.05)	3.25 (4.05)	3.06 (2.00)	28.55 (10.31)	31.87 (14.50)	32.18 (14.64)	11.94 (5.39)	12.58 (5.62)	11.71 (4.61)	1.06 (0.26)	1.19 (0.31)	1.36 (0.52)
**TUG A-B**			2.14 (0.88)	2.13 (0.89)	2.61 (1.25)	40.67 (14.15)	31.84 (12.96)	34.16 (15.40)	20.87 (9.12)	15.95 (7.47)	14.24 (6.61)	1.17 (0.29)	1.98 (0.87)	1.68 (0.52)
**TUG B-C**			2.74 (0.90)	2.77 (0.88)	2.79 (1.35)	10.32 (3.38)	19.89 (7.25)	17.89 (7.16)	4.08 (1.78)	7.87 (4.25)	7.74 (4.69)	0.51 (0.16)	1.29 (0.55)	1.51 (1.12)
**TUG C-D**			2.76 (1.10)	2.62 (0.87)	3.28 (1.84)	20.34 (10.36)	29.12 (21.47)	28.09 (30.96)	7.88 (4.18)	11.72 (11.36)	11.11 (18.64)	0.51 (0.13)	2.21 (1.39)	2.37 (1.43)
**TUG D-E**			2.69 (0.95)	2.75 (0.95)	5.80 (10.01)	43.45 (12.83)	37.18 (15.31)	46.55 (20.90)	17.92 (7.10)	15.33 (7.35)	12.88 (7.38)	1.09 (0.22)	1.77 (0.59)	2.49 (1.78)

**Table 3 sensors-20-00689-t003:** Internal validity, reliability, and responsiveness outcomes from the variables extracted from the RGB-D camera. Inertial measurement unit (IMU); CAM. RGB-D camera; lie-to-sit (LTS); sit-to-stand (STS); timed up-and-go (TUG); area under the curve (AUC); intraclass correlation coefficient (ICC).

		r IMU-CAM	ICC CAM	SEM CAM	AUC CAM
**Stairs**	*Time(s)*	0.99	0.85 (0.75–0.92)	0.46	0.60 (0.37–0.80)
	*Displacement (°)*	0.17	0.42 (0.20–0.64)	6.65	0.77 (0.55–0.91)
	*Velocity (°/s)*	0.14	0.33 (0.11–0.57)	2.25	0.84 (0.63–0.96)
	*Acceleration (°/s^2^)*	0.11	0.46 (0.23–0.66)	0.19	0.71 (0.48–0.87)
**Bending**	*Time(s)*	0.99	0.93 (0.88–0.96)	0.46	0.84 (0.63–0.96)
	*Displacement (°)*	0.58	0.75 (0.60–0.86)	11.08	0.55 (0.33–0.76)
	*Velocity (°/s)*	0.80	0.83 (0.71–0.91)	6.38	0.78 (0.56–0.92)
	*Acceleration (°/s^2^)*	0.53	0.83 (0.71–0.90)	0.17	0.84 (0.63–0.96)
**Reaching**	*Time(s)*	0.99	0.76 (0.61–0.86)	0.51	0.71 (0.48–0.88)
	*Displacement (°)*	0.20	0.37 (0.14–0.59)	2.83	0.53 (0.31–0.74)
	*Velocity (°/s)*	0.35	0.58 (0.38–0.75)	1.50	0.58 (0.34–0.76)
	*Acceleration (°/s^2^)*	0.28	0.71 (0.55–0.84)	0.13	0.56 (0.34–0.76)
**Sock**	*Time(s)*	0.99	0.72 (0.55–0.84)	1.14	0.66 (0.43–0.84)
	*Displacement (°)*	0.53	0.73 (0.57–0.85)	12.49	0.55 (0.33–0.75)
	*Velocity (°/s)*	0.55	0.83 (0.71–0.91)	2.94	0.55 (0.33–0.76)
	*Acceleration (°/s^2^)*	0.61	0.64 (0.45–0.79)	0.26	0.56 (0.33–0.76)
**LTS**	*Time(s)*	0.98	0.62 (0.41–0.79)	1.20	0.56 (0.32–0.79)
	*Displacement (°)*	0.11	0.48 (0.25–0.70)	27.08	0.56 (0.32–0.78)
	*Velocity (°/s)*	0.24	0.39 (0.15–0.63)	6.22	0.58 (0.34–0.80)
	*Acceleration(°/s^2^)*	0.09	0.16 (−0.06–0.42)	1.01	0.69 (0.44–0.88)
**STS**	*Time(s)*	1.00	0.92 (0.85–0.95)	0.41	0.85 (0.64–0.96)
	*Displacement (°)*	0.59	0.73 (0.56–0.85)	7.67	0.64 (0.42–0.85)
	*Velocity (°/s)*	0.73	0.75 (0.59–0.86)	3.07	0.72 (0.50–0.89)
	*Acceleration (°/s^2^)*	0.59	0.64 (0.45–0.79)	0.26	0.77 (0.55–0.91)
**TUG A–B**	*Time(s)*	0.99	0.90 (0.82–0.95)	0.27	0.70 (0.48–0.87)
	*Displacement (°)*	0.15	0.44 (0.21–0.65)	10.15	0.51 (0.30–0.72)
	*Velocity (°/s)*	0.36	0.41 (0.18–0.62)	5.65	0.72 (0.49–0.88)
	*Acceleration (°/s^2^)*	0.66	0.57 (0.36–0.75)	0.44	0.75 (0.53–0.91)
**TUG B–C**	*Time(s)*	0.93	0.86 (0.77–0.93)	0.36	0.75 (0.53–0.90)
	*Displacement (°)*	0.20	−0.01 (−0.19–0.23)	21.73	0.64 (0.41–0.83)
	*Velocity (°/s)*	0.51	0.21 (−0.01–0.46)	4.67	0.75 (0.52–0.90)
	*Acceleration (°/s^2^)*	0.60	0.08 (−0.11–0.33)	0.60	0.51 (0.29–0.72)
**TUG C–D**	*Time(s)*	0.95	0.80 (0.67–0.89)	0.49	0.74 (0.51–0.90)
	*Displacement (°)*	0.06	0.49 (0.27–0.69)	22.92	0.63 (0.40–0.81)
	*Velocity (°/s)*	−0.12	0.54 (0.33–0.73)	7.35	0.55 (0.33–0.75)
	*Acceleration (°/s^2^)*	0.16	0.35 (0.12–0.58)	1.47	0.57 (0.35–0.77)
**TUG D–E**	*Time(s)*	0.99	0.92 (0.86–0.95)	0.36	0.78 (0.56–0.92)
	*Displacement (°)*	0.52	0.43 (0.20–0.64)	12.74	0.54 (0.32–0.75)
	*Velocity (°/s)*	0.55	0.53 (0.31–0.71)	4.71	0.78 (0.57–0.92)
	*Acceleration (°/s^2^)*	0.38	0.30 (0.08–0.542)	0.88	0.71 (0.48–0.87)

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
