# Peer review of "Validation, Reliability, and Responsiveness Outcomes of Kinematic Assessment with an RGB-D Camera to Analyze Movement in Subacute and Chronic Low Back Pain"

_sensors, 2020, doi:10.3390/s20030689_

Round 1
Reviewer 1 Report
This manuscript describes psychometric properties of RGB-D camera assessment of trunk movement and compares them to an IMU unit and to clinical patient reported questionnaires in patients with low back pain.
I have some specific comments:
Abstract section
Consider using "measurements" instead of "outcomes".
Episode-please rewrite as "episode of low back pain".
Results: consider using "variable results" instead of "irregular results".
Conclusion: Rewrite. Consider "Functional tasks such as bending, sit-to-stand, and sock assessed with the RGB-D camera revealed acceptable validity, reliability and responsiveness in the assessment of patients with LBP, when compared with measurements made by an IMU."
Introduction section
Please rewrite paragraph 2 and 3 so as to better clarify the reasons for doing the study.
Please write the study objectives in past tense.
Methods section
2.6: Was the distance 2m or 3m as shown in the figure?
2.12: Please rewrite 1st sentence. Please remove 'the' before 'fatigue'. Please replace 'obtained with' with 'calculated as'.
Discussion section
This section needs to shortened considerably.
Automated measurement was not studied in this project; therefore, any discussion of automation of measurement could be deleted.
Author Response
Dear Reviewer,
Please, find a revision of our manuscript.
We would like to thank the Reviewers for their thoughtful and constructive comments. We have considered all suggestions, and have incorporated them into the revised manuscript. We believe our manuscript is stronger as a result of the modifications. An itemized point-by-point response to the Reviewers’ comments is presented below.

Reviewer 2 Report
The paper proposes a non-intrusive motion analysis system. The performance claims are backed up with thorough experiments.
Major comments:
1. The authors may have heard about 3D optical motion analysis system, e.g., VICON system https://www.vicon.com/. It requires retro-reflective markers attached to the subject, but is of high accuracy. To strengthen the motivation of the paper, it would be better to add some discussion about why using RGB-D camera is preferred compared to inertial measurement system (already did) and 3D motion analysis system.
2. There are some other works very similar to the authors as well. Specifically, the authors may consider adding the following references:
M. Ye, C. Yang, V. Stankovic, L. Stankovic and A. Kerr, "A Depth Camera Motion Analysis Framework for Tele-rehabilitation: Motion Capture and Person-Centric Kinematics Analysis," in IEEE Journal of Selected Topics in Signal Processing, vol. 10, no. 5, pp. 877-887, Aug. 2016.
Cheng Yang, Ukadike C. Ugbolue, Andrew Kerr, et al., “Autonomous Gait Event Detection with Portable Single-Camera Gait Kinematics Analysis System,” Journal of Sensors, vol. 2016, Article ID 5036857, 8 pages, 2016.
Author Response

(The authors gave the same response as above.)

Round 2
Reviewer 1 Report
The revised manuscript is improved compared with the original submission. However, there remain several lines in the manuscript that need to be revised to make it more readable, and clearer to the reader.
Some examples (not a comprehensive list) include
Introduction: Sentence starting with "The information..." is unclear and needs to be revised.
Sentence starting with "Depth cameras are easy....." could end with "...kinematic studies, and can easily translate from the lab to everyday clinical environments".
Remove "the" before "low back pain", before "implementation" and before "poor correlation"
Statistical analysis: Rewrite first line-Suggestion: "The internal validity....responsiveness were analyzed using the previously outlined variables".
Replace "will be" with "was".
Discussion:
Rewrite first 2 lines under "Six functional tests".
Rewrite rest of the paragraph starting with "Analyzing the observed results..."
Rewrite sentence starting with "Regarding the sock test..."
Author Response
Dear Reviewer,
Please, find a revision of our manuscript.
We would like to thank the Reviewer for their thoughtful and constructive comments. We have considered all suggestions, and have incorporated them into the revised manuscript. We believe our manuscript is stronger as a result of the modifications. An itemized point-by-point response to the Reviewer's comments is presented below.
